# Routine EWS Fusion Analysis in the Oncology Clinic to Identify Cancer-Specific Peptide Sequence Patterns That Span Breakpoints in Ewing Sarcoma and DSRCT

**DOI:** 10.3390/cancers15051623

**Published:** 2023-03-06

**Authors:** Peter M. Anderson, Zheng Jin Tu, Scott E. Kilpatrick, Matteo Trucco, Rabi Hanna, Timothy Chan

**Affiliations:** 1Pediatric Hematology and Bone Marrow Transplant, Pediatric and Taussig Cancer Institutes, Cleveland Clinic, Cleveland, OH 44195, USA; 2Bioinformatics, Molecular Pathology and Cytogenomics, Department of Laboratory Medicine, Pathology and Laboratory Medicine Institute, Cleveland Clinic, Cleveland, OH 44195, USA; 3Orthopedic Pathology and Center for ePathology, Pathology and Laboratory Medicine Institute, Cleveland Clinic, Cleveland, OH 44195, USA; 4Center for Immuno-Oncology, Department of Radiation Oncology, Taussig Cancer Institute and Lerner Research Institute, Cleveland Clinic, Cleveland, OH 44195, USA

**Keywords:** fusion gene analysis, cancer-specific sequence, EWS-FLI1, EWS-ERG, EWS-WT1, mRNA, polypeptide, breakpoint, next generation sequencing (NGS), information and education virtual visit

## Abstract

**Simple Summary:**

EWS-based fusions are aberrantly fused genes that drive Ewing sarcoma and desmoplastic small round cell tumor (DSRCT). Fusion data may be useful for personalized mRNA vaccines but are not yet routinely obtained in clinical practice. We present our workflow for the characterization of EWS driver fusions in a real-world pediatric oncology setting. We use rapid targeted sequencing of the breakpoint and genetic analysis to determine fusion sequences. We report amino acid fusion sequences from the EWS gene and the fusion partner gene (FLI1, ERG, FEV, WT1). Our workflow allows easy discernment of clinically relevant similarities and differences of EWS fusions. This simple analysis allows an understanding of molecular features of driver fusions underlying Ewing sarcoma or DSRCT in a real-world setting. This workflow is being utilized to obtain fusion neoantigen data used in “personalized” cancer vaccine trials under designs that seek to stimulate an immune response against Ewing sarcoma or DSRCT.

**Abstract:**

(1) Background: EWS fusion genes are associated with Ewing sarcoma and other Ewing family tumors including desmoplastic small round tumor, DSRCT. We utilize a clinical genomics workflow to reveal real-world frequencies of EWS fusion events, cataloging events that are similar, or divergent at the EWS breakpoint. (2) Methods: EWS fusion events from our next-generation sequencing panel (NGS) samples were first sorted by breakpoint or fusion junctions to map out the frequency of breakpoints. Fusion results were illustrated as in-frame fusion peptides involving EWS and a partner gene. (3) Results: From 2471 patient pool samples for fusion analysis at the Cleveland Clinic Molecular Pathology Laboratory, we identified 182 fusion samples evolved with the EWS gene. They are clustered in several breakpoints: chr22:29683123 (65.9%), and chr22:29688595 (2.7%). About 3/4 of Ewing sarcoma and DSRCT tumors have an identical EWS breakpoint motif at Exon 7 (SQQSSSYGQQ-) fused to a specific part of FLI1 (NPSYDSVRRG or-SSLLAYNTSS), ERG (NLPYEPPRRS), FEV (NPVGDGLFKD) or WT1 (SEKPYQCDFK). Our method also worked with Caris transcriptome data, too. Our primary clinical utility is to use this information to identify neoantigens for therapeutic purposes. (4) Conclusions and future perspectives: our method allows interpretation of what peptides result from the in-frame translation of EWS fusion junctions. These sequences, coupled with HLA-peptide binding data, are used to identify potential sequences of cancer-specific immunogenic peptides for Ewing sarcoma or DSRCT patients. This information may also be useful for immune monitoring (e.g., circulating T-cells with fusion-peptide specificity) to detect vaccine candidates, responses, or residual disease.

## 1. Introduction

Ewing sarcoma treatment has become relatively standardized with the vincristine, dexrazoxane + doxorubicin, and cyclophosphamide + mesna cycles alternating with ifosfamide + mesna and etoposide cycles [1,2,3]. Dose intensification using higher doses of chemotherapy or consolidation with high-dose chemotherapy and autologous transplant has been performed with no or modest improvement, respectively [4,5,6,7,8]. Despite apparent immunogenicity in a study of EWS-FLI1 consolidative immunotherapy [9], Ewing sarcoma vaccines against EWS peptides or mRNA coding for EWS peptides have not become part of standard-of-care or are not in clinical trials yet. Patients with excellent (100%) necrosis have a much better prognosis; Ewing sarcoma patients with pelvic tumors or poor chemotherapy response have a poor prognosis [10,11,12,13]. Similarly, poor necrosis, axial location, relapsed, and/or metastases continue to have poor survival [11,12,13,14,15,16]. How the EWS-fusion genes translate into biologic behavior is an active area of research (this issue of *Cancers,* Ewing Sarcoma: Basic Biology, Clinical Challenges and Future Perspectives, and references [17,18,19,20,21,22,23,24,25,26,27,28,29,30]).

In our pediatric and medical oncology practice, we see many high-risk and metastatic Ewing sarcoma and DSRCT patients, both in person and also using informational/educational virtual visits. Our pathologists also have extensive experience in diagnosing bone and soft tissue sarcomas, including the use of an in-house sarcoma NGS panel (refs. [31,32,33,34], Figure 1) and the use of whole exome RNA sequencing reports from Caris since January 2022. An unmet need exists to not just diagnose Ewing sarcoma and DSRCT using molecular tools but also to learn more about the nature of the EWS fusion genes and potential best targets in the context of HLA in each Ewing sarcoma or DSRCT patient [9,28,29,30]. Collecting real-world information about EWS gene fusions and resulting peptides may be leveraged to help design future immunologic approaches including identifying in a more precise manner (e.g., with HLA binding assays, T-call and B-cell recognition assays) what constructs to include in personalized cancer vaccines. The first step is to understand which cancer-specific peptides (neoantigens) are present in a real-world clinical practice setting and then seek to routinely obtain more information (e.g., HLA type). The HLA type is now routinely available on Caris reports. Then, we can better understand similarities and differences between how patients –may possibly stimulate the most effective T-cell and B-cell responses to EWS neoantigens.

Different breakpoints, or “types”, in the fusion genes have been identified for some time now, and their prognostic significance has been speculated upon previously [35]. With the routine availability in our clinical practice of NGS and transcriptome sequencing panels (e.g., Caris) as part of sarcoma pathology and molecular reports detailing the presence of fusion genes, we have developed practical tools to investigate whether identical, similar, or very different peptides are made from the EWS-partner fusion genes in Ewing sarcoma and DSRCT in our clinical population. Our team is also interested in immunogenicity analysis of EWS fusion peptides and other cancer-specific peptides (neoantigens) to use as neoantigen targets for new therapies [36,37]. From this perspective, we share current efforts and data on EWS-FLI1, EWS-ERG, EWS-FEV, and EWS-WT1 gene fusions from our Cleveland Clinic NGS sarcoma panel as well as Caris transcriptome data. Thus, we are at an early stage of using this information to facilitate ongoing analysis and prediction of the immunogenicity of EWS fusion peptides in the context of HLA. This information may help to overcome barriers to effective new immunologic approaches against cancer-specific neoantigens in Ewing sarcoma and DSRCT [36,37,38,39].

## 2. Materials and Methods

Pathology reports and gene fusions were analyzed using the Cleveland Clinic NGS sarcoma panel from January 2019. Since January 2022, we also have available data from Caris reports for transcriptome analysis. The Cleveland Clinic sarcoma NGS panel is based on anchored multiplex polymerase chain reaction (PCR) enriched for 34 gene targets [32]. The amplicons were subjected to massively parallel sequencing with 151x2 cycle pair-end reads. An informatics pipeline was used for read alignment (GRCh37 as reference genome), fusion identification, and annotation. For in-house NGS or Caris analysis, the sequenced short reads were aligned to the reference genome (hg19 or GRCh37) by either Archer Analysis or Caris transcriptome analysis pipeline. To obtain DNA and peptide sequences around the fusion junction, result alignment bam file (s) were loaded into Integrative Genome Viewer (IGV). Breakpoints were manually examined, and DNA sequences were extracted. This sequence was then translated to a peptide sequence. For ease of comparison, both fusion partner DNA and peptide sequences were represented with a different color to identify the exact point of fusion. Figure 2 shows EWS genomic (hg19) breakpoint distribution.

This in-house reporting system was used to display data as the DNA sequence spanning the fusion gene breakpoint and the amino acid sequence of an in-frame peptide spanning the gene fusion breakpoint. For ease of interpretation, the data was displayed to allow “at-a-glance” pattern recognition of identical, similar, or unique EWS-FLI1, EWS-ERG, EWS-FEV, and EWS-WT1 gene fusions (Figure 3) through visual examine fusion data via Integrative Genome Viewer [40]. DNA sequences across the gene fusion breakpoint and translational amino acid sequences were compared in Ewing sarcoma and DSRCT patients seen in the clinic and during virtual visits [33,34] who had provided paraffin blocks of tumor samples for analysis using our in-house NGS and/or Caris analysis. We also had both In-house NGS using Archer and Caris transcriptome using STARfusion data compared in 18 patients [EWS-FLI1 (*n* = 14), EWS-WT1 (*n* = 3), and EWS-FEV (*n* = 1)] This allowed us to demonstrate that the generalization of our methods pertain not only out NGS but other platforms that can help oncologists and others know “at-a-glance” similarities and difference in individual EWS gene fusions.

## 3. Results

From 190 identified fusion samples with the EWS gene, the breakpoints in a genomic location are plotted in Figure 2. Most of the fusions are clustered in several breakpoints including chr22:29683123 (65.9%), chr22:29684775 (9.3%), chr22:29688158 (8.2%), chr22: 29692358 (4.9%), and chr22:29688595 (2.7%), (Figure 2). Additional comparison data using Caris transcriptome was available for EWS-FLI1 (*n* = 14), EWS-WT1 (*n* = 3), and EWS-FEV (*n* = 1).

A typical analysis is depicted in Figure 3. The presence of start (green) or stop (red) codons at the bottom of Figure 3 shows out-of-frame sequences compared to read-through regions for the in-frame sequences. Table 1 shows the frequencies and sequence details of the most common 24 amino acid polypeptides that span the EWS breakpoints in EWS-FLI1 fusion. Table 2 details the in-frame sequence of EWS-ERG, and EWS-WT1 fusions. In-frame analysis showed the EWS motif from exon 7 (SQQSSSYGQQ) accounted for over 80% of gene fusions involving EWS-FLI1, EWS-ERG, EWS-FEV, and EWS WT1, while the remaining gene fusions involved other parts of the EWS gene.

In a real-world clinical practice (PA), using in-frame analysis of EWS-FLI1 peptides that span the breakpoint showed the EWS breakpoint sequence SQQSSSYGQQ to be the most common in our analysis (Table 1). The most common FLI1 sequence was NPSYDSVRRG. Some samples yielded an aspartate (D) instead of asparagine (N) at the FLI1 fusion point, but on manual inspection, this is related to breakpoint yielding asparagine. Although many EWS-FLI1 gene fusion events are possible, it seems that a limited number occur. Furthermore, the majority have identical breakpoints as identified by in-frame peptide analysis. Using Caris transcriptome data, we confirmed that our method is a general one as depicted in Figure 4.

Similarly, for EWS-ERG, EWS-FEV, and EWS-WT1, gene fusion analysis using in-frame peptide sequences, a common EWS sequence in EWS-ERG, EWS-FEV, and EWS-WT1 fusions was also SQQSSSYGQQ (Table 2). Thus, although many EWS-ERG, EWS-FEV, and EWS-WT1 gene fusion events are possible, it appears a few are more frequently associated with the development of Ewing sarcoma and DSRCT, respectively.

## 4. Discussion

EWS fusion genes are critical mutations associated with Ewing sarcoma and DSRCT. How these events translate into the malignant phenotype has and continues to be an area of very active investigation with many downstream effects resulting from the action of the EWS fusion genes [9,18,19,22,23,24,25,28,29,30,41,42,43,44,45]. We sought to try to learn whether EWS-FLI1, EWS-ERG, EWS-FEV, and EWS-WT1 gene fusions would result in identical, similar, or very different polypeptides using in-frame analysis. It seems that our method identified identical polypeptides for a particular fusion event (e.g., EWS–FLI1 exon 7-7). As expected, some types of fusions were more common than others (Table 1 and Table 2). In particular, the EWS gene breaks and fuses to a partner in a way to often yields SQQSSSYGQQ fused to a partner sequence for both Ewing sarcoma and DSRCT in 83% of cases. Nevertheless, both Ewing sarcoma and DSRCT can result from a variety of different fusions involving EWS and a partner gene. Thus, a unifying pathway of fusion gene action resulting in protein action (s) may be one of the numerous actions of fusion genes and/or polypeptides leading to the Ewing sarcoma and DSRCT cancer phenotypes (many paths lead to Rome analogy). Since effective epitopes for B cells may be different from T cells as reported by Liu et al. [28], precise knowledge of sequences that span EWS breakpoints to augment the additional study of best neoantigens will be important for both humoral as well as T-cell responses to neoantigens.

Our in-frame analysis could have several important future clinical applications including (1) identification of different sequences and lengths of peptides that span the specific breakpoint to test for patient-specific HLA binding to the cancer-specific peptide (e.g., selection of best neoantigens for personalized cancer vaccines); (2) analysis of T-cell and B-cell immune responses prior to immunization, during therapy, after therapy, and in response to chemotherapy, radiation, cryoablation, or immunologic interventions. These could include immune-stimulatory drugs, reduction in the tumor inhibitory microenvironment (e.g., inflammation reduction, stereotactic body radiotherapy (SBRT), freezing of tumors, and depletion of regulatory t-cells [25,46,47,48]). This information could also increase the success of priming and boosting the immune system with vaccines composed of fusion peptides mRNA for the unique fusion gene sequence in the context of a particular HLA type and neoantigen binding (personalized mRNA vaccines using precision data). Furthermore, using Caris transcriptome data, we showed that our method has general applicability for others interested in helping physicians, genetic counselors, patients, and caregivers to increase their understanding about cancer-specific sequences resulting from somatic genetic events in Ewing sarcoma and DSRCT in the oncology clinic. Thus, we are in at an early stage in the process of starting to obtain more personal and precise data for better future interventions, the next generation of evidenced based medicine in oncology as recently described by Subbiah [49].

Not only gene fusions, but also frameshift mutations create neoantigens that start with a normal sequence and then transition to an unexpected cancer-specific sequence as part of early or late mutational events in cancer. Thus, our method may have applications not only for gene fusions such as EWS-FLII1, EWS-ERG, EWS-FEV, and EWS- WT1 but also for other gene fusions which are commonly found in sarcomas and also somatic frameshift mutations. Since the best choice of neoantigens in the context of HLA is vital, the collection of both in-frame fusion peptide data as well as HLA type may someday become a more routine Ewing sarcoma and DSRCT practice as part of tumor and host testing to f additional immunologic interventions in higher-risk, metastatic, and relapsed Ewing sarcoma, DSRCT, and other solid tumor patents (such as fusion-related sarcomas). Since acid decalcification does not allow for DNA analysis, this is one barrier to general applications for bone sarcomas such as Ewing sarcoma. Thus, we are currently in the process of implementing improved methods of tissue fixation (e.g., EDTA) that will facilitate both pathologic and genetic analysis of bone biopsies and resected samples. This should facilitate future studies in not only Ewing sarcoma but also osteosarcoma and analysis of samples from bone metastases, too.

## 5. Conclusions

Our current sarcoma NGS panel as well as Caris transcriptome data was suitable for learning additional basic biology about EWS fusion genes including the exact breakpoint and intra- and intergroup similarities and differences involving EWS-FLI1, EWS-ERG, EWS-FEV, and EWS-WT1 in-frame polypeptides predicted by the fusion genes. Furthermore, our method of analysis also has possible future applications including analysis of immunogenicity of fusion peptides and cancer-specific genes such as frameshift mutations, point mutations, insertions, and deletions to yield cancer-specific neoantigens in the context of a patient’s HLA type. Although at an early stage, our paradigm of obtaining in-frame information may then facilitate the better design of more personal, precise, and effective data-driven immunologic interventions against Ewing sarcoma and DSRCT and other solid tumors including designing mRNA vaccines using the sequences and peptide size and HLA binding affinities with the highest chances of success [38,49].

## 6. Patents

The TC and Cleveland Clinic Foundation has a patent pending for the prediction of cancer-specific neoantigens such as EWS fusion peptides to HLA.

## Figures and Tables

**Figure 1 cancers-15-01623-f001:**
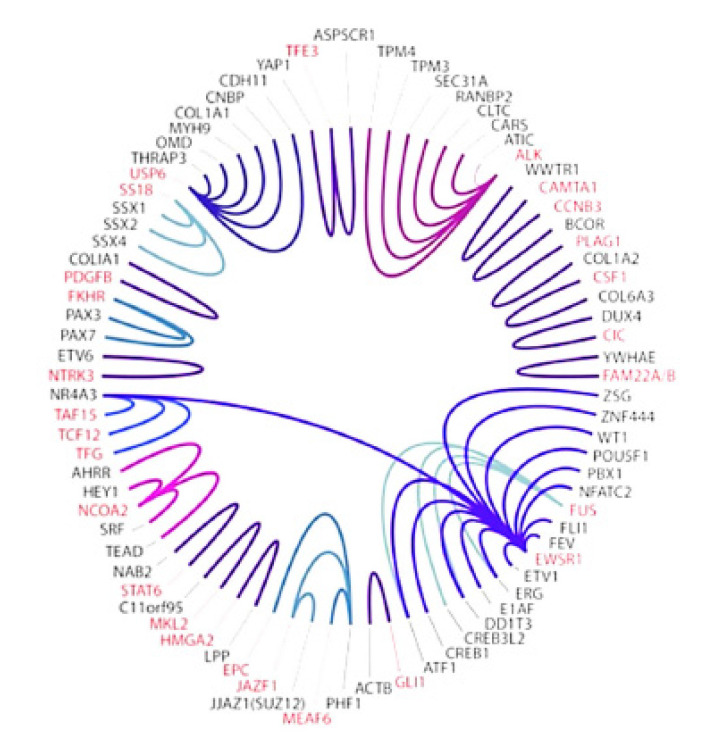
Current sarcoma NGS panel at Cleveland Clinic detects many gene fusions (same color) including 14 involving the EWS gene (Purple at 4 o’clock) [32]. Clinical NGS reports detail which exons are involved in a fusion gene such as EWS-FI1, EWS-ERG, and EWS-WT1. Since January 2022 Caris transcriptome reports are also obtained for additional information.

**Figure 2 cancers-15-01623-f002:**
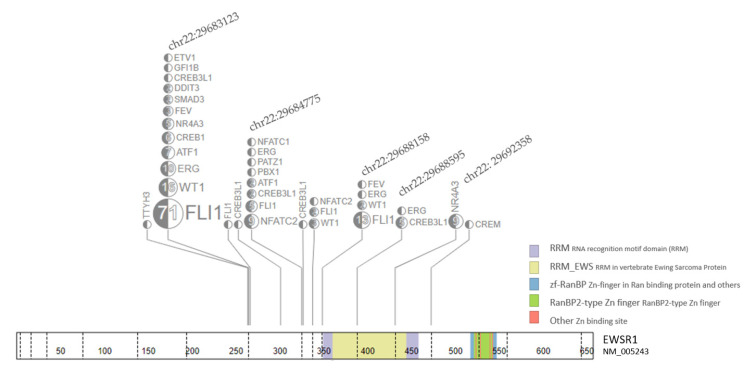
EWS fusion genomic (hg19) breakpoint distribution from 190 Fusion samples. Each breakpoints on the genomic location with samples/total are labeled on the EWS gene. The top five locations are further annotated with fusion gene in the box linked with dash line.

**Figure 3 cancers-15-01623-f003:**
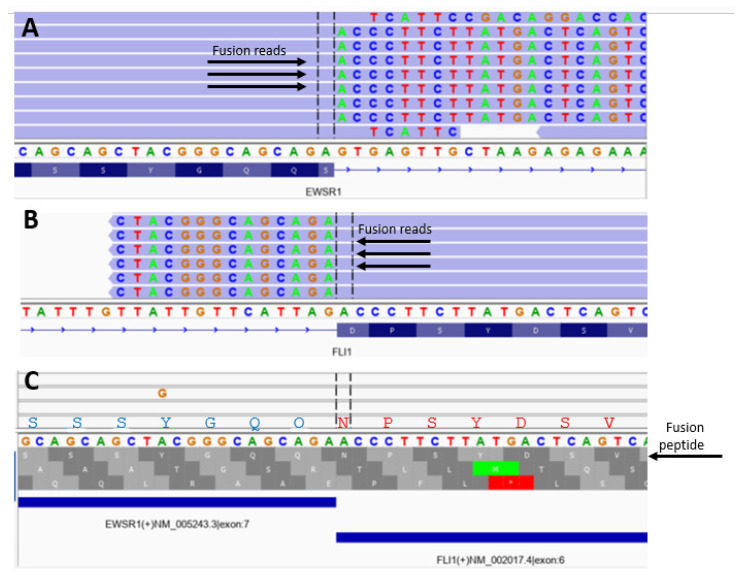
Determination of breakpoint and in-frame analysis to determine polypeptide sequences corresponding to fusion gene products in a single patient with Ewing sarcoma. (**A**) EWS gene IGV view against hg19 reference genome; (**B**) FLI1 gene IGV view against hg19 reference genome; (**C**) fusion reads IGV view against fusion contig reference. The first translational frame is the in-frame fusion with peptide sequence of SSSYGQQNPSYDSV between EWS and FLI1 gene.

**Figure 4 cancers-15-01623-f004:**
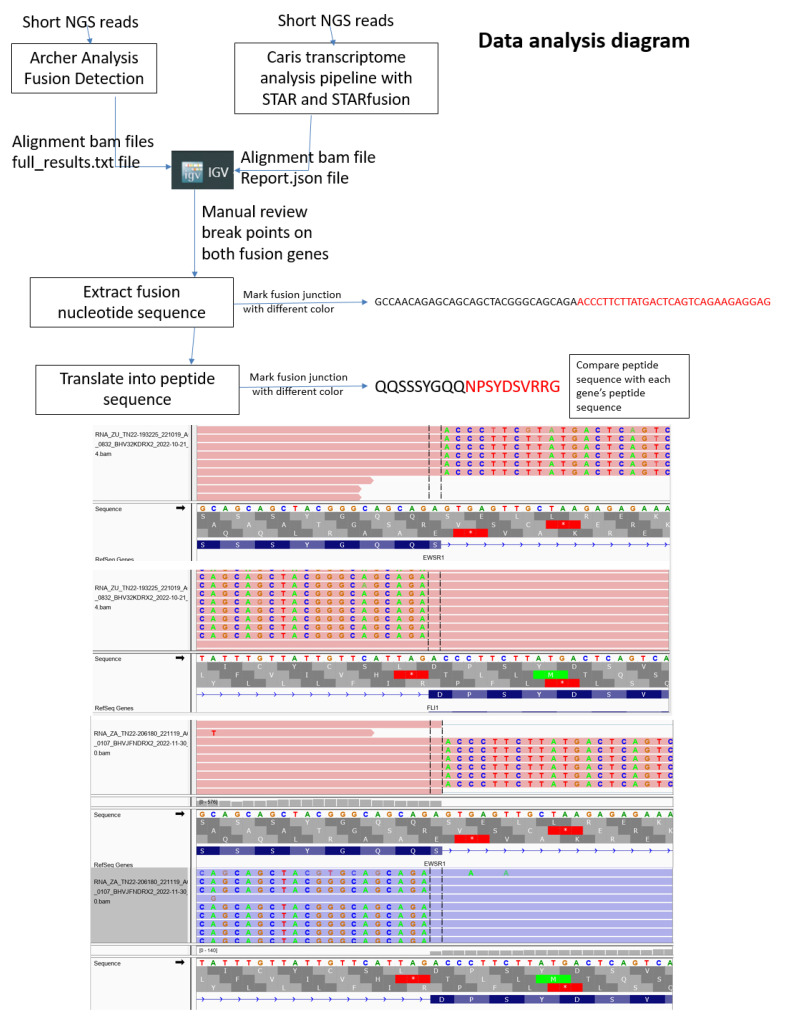
Data analysis diagram of how to depict EWS fusion junction breakpoints and corresponding amino acids. Top: flow diagram of how genomic data is depicted for in-house NGS using Archer Analysis or Caris transcriptome with STAR fusion detection. Middle and bottom: two examples of in-frame analysis of Caris transcriptome data to yield fusion peptide data with manual review of breakpoints of both fusion genes.

**Table 1 cancers-15-01623-t001:** Frequencies of Common EWS-FLI1 Fusion genes and corresponding polypeptides that span the EWS breakpoint.

EWS-FLI1 Breakpoint Fusion Peptide Analysis	Number (Per Cent)
SQQSSSYGQQ NPSYDSVRRG	19
SQQSSSYGQQ SSLLAYNTTS	12
SQQSSSYGQQ NPYQILGPTS	1
SQQSSSYGQQ RSGQIQLWQF	1
	33/40 (83%)
Other EWS-FLI1 Fusions Peptides	
PMDEGPDLDL GSLLAYNTTS	5
GERGGFNKPG GPPLGGAQTI	1
CVEFSSLIDQ PVYPDVLASG	1
	7/40 (17%)

**Table 2 cancers-15-01623-t002:** EWS-ERG, EWS-FEV, and EWS-WT1 sequences and corresponding fusion polypeptides that span the breakpoints in Ewing sarcoma and DSRCT.

Ewing Sarcomas with EWS-ERG Fusions
EWS-ERG exon 7-9 (N = 3/6, 50%) SQQSSSYGQQNLPYEPPRRS
Ewing sarcomas with EWS-FEV fusions
EWS-FEV1 exon 7-2 (2/3; 67%) SQQSSSYGQQNPVGDGLFKD
DSRCT with EWS-WT1 fusions
EWS-WT1 exon 7-7 (N = 9/12, 75%) SQQSSSYGQQSEKPYQCDFK
EWS-WT1 exon 9-7 (N = 3/12 25%) GERGGFNKPGGEKPYQCDFK

## Data Availability

Data supporting reported results can be obtained by contacting P.M.A. and Z.J.T. Zheng Jin Tu can provide additional details about in-frame analysis upon request (tuz@ccf.org).

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
