# Peer review of "Routine EWS Fusion Analysis in the Oncology Clinic to Identify Cancer-Specific Peptide Sequence Patterns That Span Breakpoints in Ewing Sarcoma and DSRCT"

_cancers, 2023, doi:10.3390/cancers15051623_

Round 1

Reviewer 1 Report

Dear authors few comment on the paper that highlighted my conclusions. Mainly:

Observations on materials and methods:

 Line 106: "An informatics pipeline was used..." - From figure 4 it is understood that the pipeline used for the in-house samples is Archer, but reading this sentence it seems that they have developed their own, this point needs to be clarified.

 Lines 111 and 112: "For ease of comparison both fusion partner DNA and peptide sequences were a different color to identify the exact point of fusion" - I would say more "were represented with a different color", as it stands the sentence seems more appropriate to a caption than to materials and methods.

 I also noticed some more formal things:

 - Figure 2 cannot be read well, I would suggest exporting it with better quality

 -There are several typos, for example "in-house" is spelled differently on lines 124 (In house) and 196 (inhouse)

 -Part of line 187 has a different character than the rest

Overall my impression is that it should be rejected.  I understand the usefulness of the work for a future vaccine study, but as it stands it's really hard to fathom.  While it could improve with revisions, I think it would perhaps be better for the authors to incorporate the description of this methodology into future work on the identified epitopes.

Author Response

Author Responses to reviewer 1

Thank-you for review and suggestions to improve our contribution to Cancers special issue.  “For Ewing Sarcoma: Basic Biology, Clinical Challenges and Future Perspectives”.

1-Your recommendation to introduction with additional background and references must be improved

Response: this has been implemented. We also cite additional references (Liu, Mackall, and Orentas ref 28-30) concerning the issue of antigen targets for the EWS fusion genes by experts in the field.

2-Line 106> Methods are described more accurately in detail,

Response: We improved description particularly Archer pipeline for in-house and STARfusion for Caris fusion verification- as requested. We also describe this on both methods and figure 4 as requested

3-Lines 111 and 112: "For ease of comparison both fusion partner DNA and peptide sequences were a different color to identify the exact point of fusion" - I would say more "were represented with a different color", as it stands the sentence seems more appropriate to a caption than to materials and methods.

Response: this has been modified exactly as requested

4- There are several typos, for example "in-house" is spelled differently on lines 124 (In house) and 196 (inhouse) and Part of line 187 has a different character than the rest

Response: these have been corrected. Font size is now 10 throughout manuscript, too

5- “Overall my impression is that it should be rejected.  I understand the usefulness of the work for a future vaccine study, but as it stands it's really hard to fathom.  While it could improve with revisions, I think it would perhaps be better for the authors to incorporate the description of this methodology into future work on the identified epitopes.”

Response: we agree with Reviewer 1 and recognize that this contribution is a start of a much larger project incorporating methodology into future work on identified epitopes. We know this will require analysis in context of HLA binding to more precisely determine best future EWS fusion targets.  This work is was an invited contribution for “For Ewing Sarcoma: Basic Biology, Clinical Challenges and Future Perspectives”. The manuscript illustrates the complexity of rare EWS gene fusions in Ewing sarcoma and how it will not be a simple matter to develop precision selection of “one size fits all” in-frame antigens spanning EWS breakpoints- especially in the context of HLA for future immunotherapy targets of Ewing sarcoma. This is described in more detail in the discussion. Although the methodology will be used in future work, we feel it is important and appropriate to share our findings in the context of the task ahead in the special issue.  “For Ewing Sarcoma: Basic Biology, Clinical Challenges and Future Perspectives”. Thus the article provides a future perspective in keeping with the theme of the special Ewing sarcoma issue.

Reviewer 2 Report

This is a study meant to expand our current understanding of Ewing sarcoma fusion breakpoints in the hopes of generating more effective immunotherapy against the driver fusion for survival gain. 282 fusion samples from an unknown number of unique patients are analyzed by NGS and Caris transcriptome analysis with massive parallel sequencing and alignment to reference genomes with a final result of a specific peptide sequence across the breakpoint. 

This interesting report is based on the science reported by Mackall and Helman et al in 2008, where the group at NCI examined 52 highly-selected patients with translocation positive advanced (either newly diagnosed metastatic or relapsed but slow-growing or stable) Ewing sarcoma or alveolar RMS, 22 of whom dropped out due to advancing disease or patient choice, 30 of which ultimately initiated consolidative immunotherapy with autologous T cells, dendritic cells pulsed with peptides spanning identified breakpoint regions of tumor-specific translocations, influenza vaccine, and E7 to generate HLA-A2 specificity. 23 patients actually completed therapy. Different cohorts received IL-2. The interventions were well-tolerated. In that study, immune responses to the translocation breakpoint peptides occurred in 39% but only 25% of HLA-A2 positive patients developed E7-specific responses. It was reassuring that all patients generated influenza-specific immune responses even after multiple prior lines of chemotherapy. There was no difference between cohorts receiving IL-2 or if CD40L was used with the DCs, or the dose of the DCs nor the use of indinavir (so presumably all could be eliminated / lowest useful dose given aka Cohort 1). The paper demonstrates curves comparing ESFT patients who received immunotherapy versus those that did not but given the bias towards how patients who received immunotherapy were selected, these do not appear to have much significance. It does appear that this technology does work better in recurrent patients with extremely stable disease rather than metastatic patients up-front, but again the selection bias prevents one from attributing improvements in survival to the intervention. Tellingly, when the immunotherapy cohort is compared against each other, there was no difference between patients with positive or negative immune responses. Furthermore, 2 of 2 survivors enrolled up-front with ESFT on that trial had pulmonary metastatic disease, which has a biology with baseline outcomes in line with the positive outcomes reported on trial. Interestingly, there was no difference in response rate to the different tumor-derived breakpoint peptides (EF-1 (EWS/FLI-1)*SSSYGQQN/PSYDSVRRGA or EF-2 (EWS/FLI-2)*SSSYGQ/QSSLLAYNT) and the study concluded that the breakpoint peptides used as immunogens in this trial appear poorly immunogenic overall.

EF-1 from that study is the same as the most common breakpoint fusion in the current study. While the foundational experiment's deep flaws diminish excitement for this project overall, the hope is that this group will better identify more immunogenic breakpoint peptides and/or place them in a setting that has learned the lessons from Mackall et al.

A more robust paper would next test the immunogenicity of the various peptides identified and look at T and NK cell responses in vitro and in vivo.  The authors should also consider the question of whether patient-specific epitopes will be solely effective or if additional peptide sequences identified in the literature for EWSR/FLI1 should be added to the mixture, such as those for B cells (PMID: 22562156). 

Overall the work is of high scientific importance but is just the start of a much larger endeavor. 

Author Response

Author Responses to reviewer 2

Thank-you for a very nice review and suggestions to improve our contribution to Cancers special issue.  “For Ewing Sarcoma: Basic Biology, Clinical Challenges and Future Perspectives”. We have revised the manuscript to address all concerns raised in your review in an improved manuscript as follows:

Review: This is a study meant to expand our current understanding of Ewing sarcoma fusion breakpoints in the hopes of generating more effective immunotherapy against the driver fusion for survival gain. 282 fusion samples from an unknown number of unique patients are analyzed by NGS and Caris transcriptome analysis with massive parallel sequencing and alignment to reference genomes with a final result of a specific peptide sequence across the breakpoint. 

This interesting report is based on the science reported by Mackall and Helman et al in 2008, where the group at NCI examined 52 highly-selected patients with translocation positive advanced (either newly diagnosed metastatic or relapsed but slow-growing or stable) Ewing sarcoma or alveolar RMS, 22 of whom dropped out due to advancing disease or patient choice, 30 of which ultimately initiated consolidative immunotherapy with autologous T cells, dendritic cells pulsed with peptides spanning identified breakpoint regions of tumor-specific translocations, influenza vaccine, and E7 to generate HLA-A2 specificity. 23 patients actually completed therapy. Different cohorts received IL-2. The interventions were well-tolerated. In that study, immune responses to the translocation breakpoint peptides occurred in 39% but only 25% of HLA-A2 positive patients developed E7-specific responses. It was reassuring that all patients generated influenza-specific immune responses even after multiple prior lines of chemotherapy. There was no difference between cohorts receiving IL-2 or if CD40L was used with the DCs, or the dose of the DCs nor the use of indinavir (so presumably all could be eliminated / lowest useful dose given aka Cohort 1). The paper demonstrates curves comparing ESFT patients who received immunotherapy versus those that did not but given the bias towards how patients who received immunotherapy were selected, these do not appear to have much significance. It does appear that this technology does work better in recurrent patients with extremely stable disease rather than metastatic patients up-front, but again the selection bias prevents one from attributing improvements in survival to the intervention. Tellingly, when the immunotherapy cohort is compared against each other, there was no difference between patients with positive or negative immune responses. Furthermore, 2 of 2 survivors enrolled up-front with ESFT on that trial had pulmonary metastatic disease, which has a biology with baseline outcomes in line with the positive outcomes reported on trial. Interestingly, there was no difference in response rate to the different tumor-derived breakpoint peptides (EF-1 (EWS/FLI-1)*SSSYGQQN/PSYDSVRRGA or EF-2 (EWS/FLI-2)*SSSYGQ/QSSLLAYNT) and the study concluded that the breakpoint peptides used as immunogens in this trial appear poorly immunogenic overall.

EF-1 from that study is the same as the most common breakpoint fusion in the current study. While the foundational experiment's deep flaws diminish excitement for this project overall, the hope is that this group will better identify more immunogenic breakpoint peptides and/or place them in a setting that has learned the lessons from Mackall et al.

  1. A more robust paper would next test the immunogenicity of the various peptides identified and look at T and NK cell responses in vitro and in vivo.  The authors should also consider the question of whether patient-specific epitopes will be solely effective or if additional peptide sequences identified in the literature for EWSR/FLI1 should be added to the mixture, such as those for B cells (PMID: 22562156). 

Author response: This is an outstandingly excellent point. Both the introduction and discussion contain language addressing this point with additional citations such as those of Liu, Mackall,  Orentas ref 28-30., and Subbiah (ref 49)) concerning the issue of antigen targets for the EWS fusion genes by experts in the field as well as need to use more data to have more evidence-based data for future oncology practice.

  1. Overall the work is of high scientific importance but is just the start of a much larger endeavor. 

Author response: the discussion now makes this point (also in response to reviewer 1). We recognize our contribution indeed is a start of a much larger project incorporating methodology into future work on identified epitopes. We know this will require analysis in context of HLA binding to more precisely determine best future EWS fusion targets.  This work is was an invited contribution for “For Ewing Sarcoma: Basic Biology, Clinical Challenges and Future Perspectives”. The manuscript illustrates the complexity of EWS gene fusions in Ewing sarcoma and how it will not be a simple matter to develop precision selection of “one size fits all” in-frame antigens spanning EWS breakpoints- especially in the context of HLA for future immunotherapy targets of Ewing sarcoma. This is described in more detail in the discussion. Although the methodology will be used in future work, we feel it is important and appropriate to share our findings in the context of the task ahead in the special issue.  “For Ewing Sarcoma: Basic Biology, Clinical Challenges and Future Perspectives”. Thus, the article provides a future perspective in keeping with the theme of the special issue.

Round 2

Reviewer 1 Report

Dear Authors,

the mascript did not provide sufficient arguments to improve our knowledge and remained similar to the previuos even if more references are reported.

To my opinion this paper should be rejected.